



# Thermodynamically admissible derivation of Biot's poroelastic equations and Gassmann's equations from conservation laws

Yury Alkhimenkov[1] and Yury Podladchikov[2]

[1]Department of Civil and Environmental Engineering, Massachusetts Institute of Technology, Cambridge, MA 02139, USA
[2]Institute of Earth Sciences, University of Lausanne, Switzerland

**Correspondence:** Yury Alkhimenkov (yalkhime@gmail.com)

**Abstract.** Gassmann's equations, formulated several decades ago, remain a cornerstone in geophysics due to their perceived exactness. However, a concise and rigorous derivation rooted in thermodynamic principles and conservation laws has been missing from the literature. Additionally, recent studies have pointed out potential logical inconsistencies in the original formulation. This paper introduces a derivation of Gassmann's equations, anchored in fundamental conservation laws and constitutive relations, ensuring their thermodynamic consistency. Alongside this, we extend the discussion to include Biot's poroelastic equations, which are widely used to describe the coupled behavior of fluid-saturated porous media under mechanical deformation. By demonstrating that Gassmann's equations are a specific case within the broader framework of Biot's theory, we further validate their relevance and applicability in geophysical contexts. Given the numerous independent rederivations and numerical verifications of these equations for diverse pore geometries, we affirm their robustness, provided the underlying assumptions are respected. To facilitate reproducibility and further exploration, symbolic Maple routines are provided for the derivations presented in this study.

## 1 Introduction

Gassmann's equations (Gassmann, 1951), developed several decades ago, stand as fundamental expressions in geophysics for analyzing the elastic properties of fluid-saturated porous media. These equations provide a means to predict the seismic velocities and mechanical behavior of such materials. However, despite their widespread use, recent studies have highlighted concerns regarding the logical consistency in the derivation of Gassmann's equations. This has sparked a demand for a more rigorous thermodynamically admissible framework, rooted in conservation laws and constitutive relations, to ensure their reliability and applicability in geophysical modeling and exploration.

This article aims to address these concerns by presenting a novel derivation of Biot's poroelastic equations and Gassmann's equations, which strictly adheres to fundamental conservation laws and thermodynamic principles. In particular, we leverage the formalism of classical non-equilibrium thermodynamics as described in Lebon et al. (2008), focusing on the interrelation of fluxes and forces, entropy production, and the thermodynamic admissibility of constitutive equations. This approach allows us to systematically derive the targeted equations while ensuring that the derived models are consistent with the second law of thermodynamics.





We demonstrate the thermodynamic admissibility of the derived equations and validate their integrity through theoretical analysis and numerical simulations. By incorporating the entropy production constraints and internal variables approach from classical non-equilibrium thermodynamics, we ensure that the derived models not only describe the macroscopic behavior accurately but also respect the microscopic interactions between phases in porous media. While the general methodology was outlined by Yarushina and Podladchikov (2015), this study specifically focuses on the rigorous derivation of Biot's poroelastic,

Gassmann's, and effective stress law equations, along with addressing concerns related to their physical validity.

The paper is organized as follows: First, essential equations of classical irreversible thermodynamics are presented, emphasizing the link between thermodynamic forces and fluxes. Next, we introduce the resulting evolution equations applicable to poro-viscoelastoplastic media. Following this, the target Biot's poroelastic, Gassmann's, and effective stress law equations are derived within this thermodynamically consistent framework. In the discussion section, we provide a detailed analysis of

the validity and applicability of Gassmann's equations, highlighting the importance of respecting thermodynamic principles in their derivation and use. To facilitate reproducibility, symbolic Maple routines are provided to verify the presented results. The routines archive (v1.0) is available from a permanent DOI repository (Zenodo) at https://doi.org/10.5281/zenodo.13942953 (last access: October 17, 2024) (Alkhimenkov and Podladchikov, 2024).

## 2  Assumptions and Scope of the Study

The following assumptions are made throughout the derivation of Biot's poroelastic and Gassmann's equations to ensure the validity of the results:

–  The material is assumed to be linearly elastic, and the strains are small, implying small fluid pressure perturbations relative to the confining stress.

–  The porous medium is considered homogeneous and isotropic.

–  The interactions between the solid and fluid phases are governed by linear constitutive laws, and the fluid flow obeys Darcy's law.

The constraint of zero dissipation (entropy production) during reversible poroelastic deformation provides an essential constraint on the poroelastic constitutive equation for porosity evolution.

–  The derivation assumes a quasi-static process, meaning inertia effects are ignored.

These assumptions provide a simplified framework for the derivation and are crucial for ensuring the thermodynamic admissibility of the results. Future work may extend these derivations to include non-linear elasticity, anisotropy, and dynamic effects.



## 3 Derivation of Gassmann's Equations

### 3.1 General Representation of Classical Irreversible Thermodynamics

Porous materials can be modeled as systems consisting of two interacting phases: a solid skeleton and a saturating fluid. These phases can exchange heat, momentum, and matter, leading to complex interactions that must be captured within the framework of classical irreversible thermodynamics (Gyarmati et al., 1970; Jou et al., 1996; Lebon et al., 2008; Yarushina and Podladchikov, 2015). Using the principles of classical non-equilibrium thermodynamics, the conservation equations governing mass, momentum, entropy, and energy for each phase are expressed in the Eulerian framework as follows:

$$\frac{\partial(\rho\phi)}{\partial t} + \nabla_j \left( \rho\phi\boldsymbol{v}_j + q_\rho^j \right) = Q_p, \tag{1}$$

$$\frac{\partial(\rho\phi\boldsymbol{v}_i)}{\partial t} + \nabla_j \left( \rho\phi\boldsymbol{v}_i\boldsymbol{v}_j + q_{\boldsymbol{v}}^{ij} \right) = Q_{v_i}, \tag{2}$$

$$\frac{\partial(\rho\phi\boldsymbol{s})}{\partial t} + \nabla_j \left( \rho\phi\boldsymbol{s}\boldsymbol{v}_j + q_{\boldsymbol{s}}^j \right) = Q_s, \tag{3}$$

$$\frac{\partial(\rho\phi\boldsymbol{e})}{\partial t} + \nabla_j \left( \rho\phi\boldsymbol{e}\boldsymbol{v}_j + q_{\boldsymbol{e}}^j \right) = Q_e, \tag{4}$$

where $\boldsymbol{v}_j$, $\boldsymbol{s}$, and $\boldsymbol{e}$ denote the velocity, specific entropy, and specific total energy per unit mass, respectively. The terms $\nabla_j$ represents the partial derivative with respect to spatial coordinates, while $q_\rho^j$, $q_{\boldsymbol{v}}^{ij}$, $q_{\boldsymbol{s}}^j$, and $q_{\boldsymbol{e}}^j$ correspond to the fluxes of mass, momentum, entropy, and energy, respectively. The terms $Q_p$, $Q_{v_i}$, $Q_s$, and $Q_e$ represent the corresponding production rates due to irreversible processes (Yarushina and Podladchikov, 2015).

### Local Entropy Production

In the context of classical non-equilibrium thermodynamics (Lebon et al., 2008), each phase within the porous medium is considered to be locally in thermodynamic equilibrium, which means that intensive variables such as temperature and chemical potential are well-defined at each point. This leads to a fundamental relationship between the infinitesimal change in specific internal energy $U$ for each phase and the corresponding changes in specific entropy $S$, specific volume $\rho$, the elastic component of porosity $\phi^e$. The local entropy production is derived from the energy balance and is given by:

$$\frac{dU}{dt} = T\frac{dS}{dt} - p\frac{d(1/\rho)}{dt} + v\frac{dv}{dt} + \mu\frac{dC}{dt} + \frac{\tau_\phi}{\rho\phi}\frac{d\phi^e}{dt}, \tag{5}$$

where $\tau_\phi$ is the thermodynamic variable (pressure) conjugated to porosity change (to be defined). $\tau_\phi$ can be viewed as analogy to pressure as conjugate variable to volume change. $\frac{d}{dt} = \frac{\partial}{\partial t} + v_i\nabla_i$ denotes the Lagrangian (material) derivative with respect to a specific phase, $\frac{d\phi^e}{dt}$ is the reversible part of the porosity change.





- $T\dfrac{dS}{dt}$: Heat stored in internal energy $U$.
- $\dfrac{p}{\rho^2}\dfrac{d\rho}{dt}$: Energy change due to volumetric deformation (Hooke's Law).
- $v\dfrac{dv}{dt}$: Newtonian mechanics (kinetic energy, e.g., $v\frac{dv}{dt}=\frac{1}{2}\frac{dv^2}{dt}$).
- $\mu\dfrac{dC}{dt}$: Energy due to changes in composition (chemical potential), which is zero in the present derivation.
- $\dfrac{\tau_\phi}{\rho\,\phi}\dfrac{d\phi^e}{dt}$: Poroelastic effects: reversible part of the energy change due to the changes in porosity. Note, that $\tau_\phi$ is not defined yet.

**Entropy Production ($TQ_s$)**

Solving the local entropy production equation for $Q_s$ and multiplying both sides by $T$, we have (for details see Appendix B):

$$TQ_s = \eta\phi\left(\frac{dv}{dx}\right)^2 + \frac{\lambda\phi}{T}\left(\frac{dT}{dx}\right)^2 + pv\frac{d\phi}{dx} + \mu Q_\rho C - vQ_v - Q_\rho G_{\text{Gibbs}} + Q_u + p\frac{d\phi}{dt} - \tau_\phi\frac{d\phi^e}{dt} \tag{6}$$

This expression represents the entropy production, which must be non-negative according to the second law of thermodynamics. This formulation, which assumes local thermodynamic equilibrium for only the solid and fluid phases, is less strict than Biot's classical assumption of a single internal energy potential for the entire two-phase system in the linear poroelastic case (Yarushina and Podladchikov, 2015).

## 3.2 Extended Thermodynamic Admissibility

Building upon the concepts from Lebon et al. (2008) and the nonlinear viscoelastoplastic framework developed by Yarushina and Podladchikov (2015), the derivation of Gassmann's and Biot's equations must satisfy the constraints of thermodynamic admissibility. Specifically, the entropy production $Q_s$ must be non-negative, and the constitutive relations must be derived in a way that ensures compliance with the second law of thermodynamics.

### 3.2.1 Thermodynamic Constraints on Fluxes and Productions

The second law of thermodynamics requires that the total entropy production of the system remains non-negative. This condition applies both to the intra-phase and inter-phase entropy production within a porous medium. Mathematically, this is expressed as:

$$\sum_{\text{phases}} Q_s = \sum_{\text{phases}} Q_s^{\text{intra}} + Q_s^{\text{inter}} \geq 0. \tag{7}$$

Here, $Q_s^{\text{intra}}$ represents the intra-phase entropy production within each phase, while $Q_s^{\text{inter}}$ accounts for the inter-phase contributions due to interactions between the solid skeleton and the fluid phase. To satisfy the second law, both components must be non-negative.



**Entropy Production and Compaction Mechanisms**

In the context of poroelasticity, the most important outcome from expression (6) is in the two terms, which describe porosity change:

$$TQ_s^{poro} = p\frac{d\phi}{dt} - \tau_\phi \frac{d\phi^e}{dt} = \sum_{\text{phases}} \left( p\frac{d\phi}{dt} - \tau_\phi \frac{d\phi^e}{dt} \right).$$  (8)

We assume that the porosity evolution can be decomposed into elastic and dissipative components, which together with the negativity of entropy production requires that inelastic porosity equation takes the form (Yarushina and Podladchikov, 2015):

$$\frac{d^s\phi}{dt} - \frac{d^s\phi^e}{dt} = -\frac{p_e}{\eta_\phi},$$  (9)

where $\phi^e$ denotes the elastic portion of porosity, $p_e = \bar{p} - p_f$ represents the effective pressure (total pressure, $\bar{p} = (1-\phi)p_s + \phi p_f$, minus fluid pressure, $p_f$, respectively), and $\eta_\phi$ stands for the effective bulk viscosity. Using the definition (9), we can rewrite expression (8) in the following form:

$$TQ_s^{poro} = \sum_{\text{phases}} \left[ (p_s - \tau_\phi^s) - (p_f - \tau_\phi^f) \right] \frac{d\phi^e}{dt},$$  (10)

In equilibrium conditions, the entropy production tends to zero, which implies that the term $\left[ (p_s - \tau_\phi^s) - (p_f - \tau_\phi^f) \right] = 0$ (!). The fluid phase does not contain the porosity term, meaning that $\tau_\phi^f = 0$. It implies that $\left[ (p_s - \tau_\phi^s) - (p_f - \tau_\phi^f) \right] = 0$ corresponds to $\tau_\phi^s = p_s - p_f$ (Yarushina and Podladchikov, 2015). We also notice that $\tau_\phi^s = p_e/(1-\phi)$. By definition the poroelastic constant $K_\phi$ is defined that as linear rheological relationship during reversible poroelastic part of deformation:

$$\frac{d\phi^e}{dt} = K_\phi(1-\phi)\frac{d\tau_\phi^s}{dt} = K_\phi \frac{dp_e}{dt},$$  (11)

The statement (11) means that changes in porosity are proportional to changes via $\tau_\phi^s$, which is the pressure difference $p_e/(1-\phi)$. Due to the requirement of zero entropy production, this statement provides us with the definition that equal changes in pressures leave porosity unchanged.

One of the key assumptions made during the original derivation of Gassmann's equations (Gassmann, 1951) is that equal changes in pore (fluid) pressure and confining (total) pressure leave the porosity unchanged. This assumption holds when considering a homogeneous elastic frame material (Korringa, 1981; Alkhimenkov, 2024). Any discrepancy in total and fluid pressure changes will lead to porosity changes as follows from equation (11). As highlighted by Korringa (1981), applying confining (external) pressure to a homogeneous elastic frame material causes it to behave as a linear mapping. **Note that in the present thermodynamically admissible model, this is not assumed, but derived as a condition necessary to ensure zero entropy production during reversible poroelastic processes**.

After simplifying and collecting terms (see Appendix B), the total entropy production becomes:

$$TQ_{s,\text{total}} = \frac{1}{\eta_\phi} \left( \frac{p_e}{(1-\phi)} \right)^2 + \eta_t \left( \text{div}\, v^s \right)^2 + \frac{(q^D)^2 \eta_{\text{dv}}}{\phi} + \frac{\lambda_t}{T} \left( \frac{\partial T}{\partial x} \right)^2$$  (12)





- $-\dfrac{1}{\eta_\phi}\left(\dfrac{p_e}{(1-\phi)}\right)^2$ : Entropy production due to poroelastic deformation (poroelastic coefficient $\eta_\phi$ and pressure difference $p_e$).

   - $\eta_t\,(\operatorname{div} v^s)^2$ : Entropy production due to viscous dissipation in the solid phase.

   - $\dfrac{(q^D)^2\eta_{\mathrm{dV}}}{\phi}$ : Entropy production due to viscous dissipation in fluid flow (Darcy flow).

   - $\dfrac{\lambda_t}{T}\left(\dfrac{\partial T}{\partial x}\right)^2$ : Entropy production due to heat conduction (Fourier's law).

**The non-negative nature of each term ensures the overall positivity of entropy production, thereby confirming the thermodynamic validity of the system**.

For detailed derivations and applications of these principles to specific pore geometries and boundary conditions, readers are encouraged to refer to Appendix A, Appendix B, and the discussions provided by Yarushina and Podladchikov (2015). Additionally, symbolic Maple routines used to reproduce and validate the theoretical results presented in this article are available in

a permanent DOI repository (Zenodo) *will be provide after review, now see suppl. material*. For a detailed explanation of the Maple script used in the derivation and analysis of entropy production in a single-phase medium, see Appendix A. Appendix B provides a similar explanation for the entropy production derivation in a two-phase porous medium.

### 3.3 Two-phase media: fluid-saturated porous material

The equations governing fluid flow in poro-viscoelastoplastic media can be formulated based on the conservation laws and

constitutive equations for both fluid and solid phases.

#### 3.3.1 Conservation of linear momentum and Darcy's law

The conservation of linear momentum is

$$\nabla_j(-\bar{p}\delta_{ij}+\bar{\tau}_{ij})-g_i\bar{\rho}=0, \tag{13}$$

where $\bar{p}=(1-\phi)p_s+\phi p_f$ is the total pressure, $\bar{\tau}_{ij}$ is the deviatoric stress tensor, $\delta_{ij}$ is the Kronecker delta, $i,j=\overline{1..3}$ and

Einstein summation convention is used (summation is applied over repeated indexes). Viscous fluid flow through porous media is governed by Darcy's law:

$$q_i^{\mathrm{D}}=-\frac{k}{\eta_f}(\nabla_i p^f+g_i\rho^f), \tag{14}$$

where $q_i^D=\phi(v_i^f-v_i^s)$ denotes Darcy's flux, $v_i^f$ denotes the fluid velocity, $v_i^s$ denotes the solid velocity, $k$ is permeability, $\eta_f$ is fluid shear viscosity.

#### 3.3.2 Conservation of mass

Conservation of mass for fluid phase is

$$\frac{\partial(\phi\rho_f)}{\partial t}+\nabla_j\left(\phi\rho_f v_j^f\right)=0, \tag{15}$$





where $\rho_f$ denotes fluid density and conservation of mass for solid phase is

$$\frac{\partial((1-\phi)\rho_s)}{\partial t} + \nabla_j\left((1-\phi)\rho_s v_j^s\right) = 0, \tag{16}$$

where $\rho_s$ denotes solid density. Equations (15)-(16) can be reformulated for divergences $\nabla_j v_j^s$ and $\nabla_j q_j^D$:

$$\nabla_j v_j^s = -\frac{1}{\rho_s}\frac{d^s\rho^s}{dt} + \frac{1}{1-\phi}\frac{d^s\phi}{dt} \tag{17}$$

and

$$\nabla_j q_j^D = -\frac{\phi}{\rho_f}\frac{d^f\rho^f}{dt} - \frac{d^s\phi}{dt} - \phi\nabla_j v_j^s, \tag{18}$$

where $\dfrac{d^s}{dt} = \dfrac{\partial}{\partial t} + v_i^s\nabla_i$ denotes the Lagrangian (material) derivative with respect to solid and $\dfrac{d^f}{dt} = \dfrac{\partial}{\partial t} + v_i^f\nabla_i$ denotes the
Lagrangian (material) derivative with respect to fluid.

### 3.3.3 Constitutive relations

Elastic compressibility for fluid and solid densities is formulated as (Yarushina and Podladchikov, 2015):

$$\frac{K_f}{\rho_f}\frac{d^f\rho_f}{dt} = \frac{d^f p_f}{dt}, \tag{19}$$

$$\frac{K_s}{\rho_s}\frac{d^s\rho_s}{dt} = \frac{1}{1-\phi}\left(\frac{d^s\bar{p}}{dt} - \phi\frac{d^f p_f}{dt}\right), \tag{20}$$

where $K_f$ denotes the fluid bulk modulus and $K_s$ denotes the solid bulk modulus. A closing relation is the equation governing porosity evolution (Maxwell viscoelastic volumetric response):

$$\frac{d^s\phi}{dt} = \frac{1}{K_\phi}\left(\frac{d^f p_f}{dt} - \frac{d^s\bar{p}}{dt}\right) + \frac{1}{\eta_\phi}(p_f - \bar{p}), \tag{21}$$

where $K_\phi$ is the poroelastic constant defined by equation (11).

### 3.3.4 Resulting evolution equations for poro-viscoelastoplastic media

By eliminating the time derivatives of densities and porosity in equations (17)-(18) using expressions (19)-(21), the following system of equations for compressibilities is obtained (Yarushina and Podladchikov, 2015):

$$\begin{pmatrix} \nabla_k v_k^s \\ \nabla_k q_k^D \end{pmatrix} = -\frac{1}{K_d}\begin{pmatrix} 1 & -\alpha \\ -\alpha & \dfrac{\alpha}{B} \end{pmatrix}\begin{pmatrix} \dfrac{d^s\bar{p}}{dt} \\ \dfrac{d^f p_f}{dt} \end{pmatrix} - \frac{1}{(1-\phi)\eta_\phi}\begin{pmatrix} 1 & -1 \\ -1 & 1 \end{pmatrix}\begin{pmatrix} \bar{p} \\ p_f \end{pmatrix}. \tag{22}$$



Deviatoric stresses are related to solid velocity gradients through the Maxwell viscoelastic relationship (Beuchert and Pod-
ladchikov, 2010):

$$\frac{1}{2G_{sat}}\frac{d^\nabla \bar\tau_{ij}}{dt} + \frac{\bar\tau_{ij}}{\eta_s} = \frac{1}{2}(\nabla_j v_i^s + \nabla_i v_j^s) - \frac{1}{3}(\nabla_k v_k^s)\delta_{ij},\tag{23}$$

where $G_{sat}$ is the shear modulus of the fluid-saturated porous material, $\dfrac{d^\nabla \bar\tau_{ij}}{dt} = \dfrac{d^s \bar\tau_{ij}}{dt} - \bar\tau_{ik}\omega_{kj} - \bar\tau_{jk}\omega_{ki}$ correspond to Jau-
mann objective stress rate and $\omega_{ki} = \dfrac{1}{2}\left(\nabla_k v_i^s - \nabla_i v_k^s\right)$ denotes the antisymmetric part of the solid velocity gradient. The
Carman–Kozeny relationship for permiability evolution as a function of porosity is

$$k = k_0 \left(\frac{\phi}{\phi_0}\right)^{n_k},\tag{24}$$

where $n_k = 3$.

### 3.4 Linear elastic limit ($\eta_\phi \to +\infty$): Biot's poroelastic equations

Under the small strain approximation and infinite $\eta_\phi$, a linear elastic limit of expression (22) can be derived which is know as
Biot's poroelastic equations (Biot, 1962):

$$\begin{pmatrix}\nabla_k v_k^s \\ \nabla_k q_k^D\end{pmatrix} = -\frac{1}{K_d}\begin{pmatrix}1 & -\alpha \\ -\alpha & \dfrac{\alpha}{B}\end{pmatrix}\begin{pmatrix}\dfrac{d\bar p}{dt} \\ \dfrac{dp_f}{dt}\end{pmatrix}.\tag{25}$$

The system of equations (25) can be rewritten for stiffness. For that let us invert the matrix of coefficients:

$$\left[\frac{1}{K_d}\begin{pmatrix}1 & -\alpha \\ -\alpha & \dfrac{\alpha}{B}\end{pmatrix}\right]^{-1} = \frac{K_d}{\alpha/B - \alpha^2}\begin{pmatrix}\dfrac{\alpha}{B} & \alpha \\ \alpha & 1\end{pmatrix} \equiv \frac{K_d}{1-\alpha B}\begin{pmatrix}1 & B \\ B & \dfrac{B}{\alpha}\end{pmatrix}.\tag{26}$$

The resulting expression for stiffness is:

$$\begin{pmatrix}\dfrac{d\bar p}{dt} \\ \dfrac{dp_f}{dt}\end{pmatrix} = -K_u\begin{pmatrix}1 & B \\ B & \dfrac{B}{\alpha}\end{pmatrix}\begin{pmatrix}\nabla_k v_k^s \\ \nabla_k q_k^D\end{pmatrix},\tag{27}$$

where $K_u = K_d\left(1-\alpha B\right)^{-1}$. Poroelastic constants in the expressions (22)-(27) are the following:

$$\alpha = 1 - \frac{K_d}{K_s}\tag{28}$$

and

$$B = \frac{1/K_d - 1/K_s}{1/K_d - 1/K_s + \phi(1/K_f - 1/K_s)}.\tag{29}$$





The relation between $K_d$, $K_s$ and $K_\phi$ (defined by equation (11)) is

$$\frac{1}{K_\phi} = \frac{1-\phi}{K_d} - \frac{1}{K_s}. \tag{30}$$

Various poroelastic constants can be calculated numerically (Alkhimenkov, 2023) or measured using physical experimentation in a laboratory (Makhnenko and Podladchikov, 2018).

### 3.5 Gassmann's equations

The relation between undrained response, $K_u$ (see expression (27) under $\nabla_k q_k^D = 0$), and drained response, $K_d$, is known as Gassmann's equation (Gassmann, 1951):

$$K_u = K_d \left(1 - \alpha B\right)^{-1}. \tag{31}$$

According to Gassmann's equations, shear modulus of a fluid-saturated rock, $G_{sat}$, is equivalent to the shear modulus of a dry rock, $G_d$ (equivalent to a drained response):

$$G_{sat} = G_d. \tag{32}$$

The expression (31) is derived from the equation (25) via inversion of matrix of coefficients leading to the expression (27). Note that English translation of the the original paper by Gassmann (Gassmann, 1951) is presented by Pelissier et al. (2007).

### 3.6 Effective stress law

Nur and Byerlee (1971) provided the exact expressions for the effective stress law, which can be treated as an exact result in poroelasticity. It is defined by the following expression (Yarushina and Podladchikov, 2015):

$$dp_{\text{eff}} = d\bar{p} - \alpha \, dp_f \equiv d\bar{p} - \left(1 - \frac{K_d}{K_s}\right) dp_f, \tag{33}$$

where $K_d$ can be measured as

$$K_d = -\frac{1}{\nabla_k v_k^s} \frac{dp_{\text{eff}}}{dt}\bigg|_{\text{undrained}}. \tag{34}$$

The exact effective stress law given by the formula (34) strictly follows from the derived expression (25).

## 4 Discussion

### 4.1 Physical Interpretation of the Derived Equations

The derived Biot's poroelastic equations describe the coupled mechanical and fluid flow behavior of a fluid-saturated porous medium. Specifically, they account for the interaction between the solid matrix deformation and the pore fluid pressure changes.



The effective stress law, which modifies the classical elastic stress by incorporating fluid pressure, plays a key role in understanding how external loads and fluid injection or extraction influence the stability and deformation of the porous medium.

Gassmann's equations provide a relation between the bulk moduli of the dry and fluid-saturated rock, offering insights into how fluid properties and porosity affect the seismic response of the material. The results show that under the assumption of quasi-static conditions and small perturbations, the derived equations capture the essential physics of wave propagation and attenuation in fluid-saturated media.

## 4.2 Derivation of Gassmann's equations and relation to poroelasticity

Gassmann's equations are directly related to the quasi-static (Biot, 1941) and dynamic poroelasticity (Biot, 1956, 1962). While the roots of the elastodynamic poroelasticity (e.g., the presence of the slow P-wave in fluid-saturated porous media) were provided by Frenkel (1944) (see also Pride and Garambois (2005)), a rigorous derivation of poroelastic equations and parameters were presented a few years later by Biot (1941); Biot and Willis (1957); Biot (1962). Many researchers have fully rederived Gassmann's equations relying on different methods (or explored specific aspects of Gassmann's equations in the framework

of poroelasticity) (Brown and Korringa, 1975; Korringa, 1981; Burridge and Keller, 1981; Zimmerman, 1990; Berryman and Milton, 1991; Berryman, 1999; Smith et al., 2003; Lopatnikov and Cheng, 2004; Gurevich, 2007; Fortin and Guéguen, 2021). Of course, a full list of scientist who contributed to poroelasticity is large, and while we acknowledge their extensive contributions, our intention in this short article is not to provide an exhaustive list. An interested reader is referred to Sevostianov (2020), which provides an extensive review of Gassmann's equations. There are several books that also might be useful, e.g.,

Bourbié et al. (1987), Zimmerman (1990), Wang (2000), Ulm and Coussy (2003), Coussy (2004, 2011), Guéguen and Boutéca (2004) Dormieux et al. (2006), Cheng (2016), Mavko et al. (2020).

### 4.2.1 Thermodynamically admissible conditions

The main assumptions behind the applicability of Gassmann's equations (21)-(32) are: (i) Linear elasticity; (ii) Small strains; (iii) Isotropic homogeneous frame material; (iv) Isotropic dry response (note that Gassmann's original publication contains an

extension to anisotropy); (v) Assumption that equal changes in pore (fluid) pressure and confining (total) pressure leave the porosity unchanged (Korringa, 1981; Alkhimenkov, 2024). Assumption (v) holds for isotropic homogeneous frame material (Korringa, 1981). In the framework of the present study, this condition is satisfied and is required for thermodynamic admissibility (see expressions (8)-(11) and the explanation therein): "The constraint of zero dissipation (entropy production) during reversible poroelastic deformation provides an essential constraint on the poroelastic constitutive equation for porosity evolu-

tion." In other words, in the present thermodynamically admissible model, (v) is not an assumption but a strict requirement for zero entropy production during reversible poroelastic processes.



### 4.3 Numerical validation of Gassmann's equations

Alkhimenkov (2023) performed a numerical validation of Gassmann's equations considering a 3D numerical setup and relatively complex pore geometry that includes narrow regions (cracks) and large pore space. Numerical calculations were performed using a finite element method and the resulting system of equations was solved using a robust direct PARDISO solver (Schenk and Gärtner, 2004). Alkhimenkov (2023) conducted a convergence study showing that, for finer resolution, the result of the numerical solution converges towards the result obtained from the original Gassmann's equation. Such a converges analysis validates the accuracy of Gassmann's equation for a particular (but arbitrary) pore geometry. Furthermore, the pore geometry that was used did not contain any special features (among all possible geometries) that were tailored to make it consistent with Gassmann's equations (Alkhimenkov, 2024). There are also other 3D numerical studies that consider different geometries of the pore space and are consistent with Gassmann's equations (Alkhimenkov et al., 2020a, b; Alkhimenkov and Quintal, 2022a, b).

### 4.4 Applicability of Gassmann's equations

Gassmann's equation (Gassmann, 1951) represented by expression (31) can be rewritten in the following form:

$$K_u = K_d + \frac{\left(1 - K_d K_s^{-1}\right)^2}{\phi K_f^{-1} + (1-\phi)K_s^{-1} - K_d/K_s^2}. \tag{35}$$

Thomsen (2023) argued that the original derivation of Gassmann's equations contains a logical error and provided an updated version of these relations (see also Brown and Korringa (1975)):

$$K_u = K_d + \frac{\left(1 - K_d K_M^{-1}\right)^2}{\phi K_f^{-1} + (1-\phi)K_s^{-1} - K_d/K_M^2}, \tag{36}$$

where $K_M$ is a new parameter, so-called "mean" incompressibility (or "mean" bulk modulus) (Thomsen, 2023). Note the similarity between expressions (35) and (36). Relation (36) contains one more parameter, $K_M$, compared to the original Gassmann's equation (35). Thomsen (2023) also provided ways to evaluate $K_M$ by using the following expressions:

$$K_M = \left[1/K_d - \frac{(1/K_d - 1/K_u)}{B}\right]^{-1}, \tag{37}$$

where $B$ is directly observable in a quasi-static experiment. Alternatively, expression (37) for $K_M$ can be exactly reformulated as:

$$K_M = \left[\frac{B\left(\phi K_f^{-1} + (1-\phi)K_s^{-1}\right) - (1-B)K_d^{-1}}{2B-1}\right]^{-1}. \tag{38}$$

Alkhimenkov (2023) conducted a numerical convergence study showing that $K_M$ is converging to $K_s$ as the resolution increases (in the numerical experiment $K_M$ was calculated independently using expression (37), so $B$ was calculated in addition to other parameters). Consequently, the result of the expression (36) is converging to the original Gassmann's formulation (35)



as the resolution increases. As a result, there is no difference between the two formulations (equations (35) and (36)) since
$K_M \equiv K_s$, that validates the original Gassmann's formulation.

We fully agree with the proposal by Thomsen (2023) that an additional measurement (or an additional parameter) can significantly improve the characterization of fluid-saturated rocks. Indeed, rocks are usually composed by several anisotropic minerals; rocks have some degree of anisotropy; rocks contain compliant cracks (or grain-to-grain contacts) and stiff pores that behave differently under loading; rocks may have some degree of heterogeneity that cannot be represented via a representative
volume element. Furthermore, the elastic moduli might be different by several percent under compression or extension. All these divergences of ideal small strain elasticity suggest more degrees of freedom and, as a consequence, more experimental (or numerical) measurements are needed to fully characterize the fully saturated realistic rocks.

## 5    Conclusions

This study has presented a novel and thermodynamically admissible derivation of both Gassmann's and Biot's poroelastic
equations, which are crucial for characterizing the elastic and coupled mechanical behavior of fluid-saturated porous media in geophysics. By adhering to conservation laws and constitutive relations, we have addressed concerns about logical inconsistencies in the original derivation of Gassmann's equations and extended the theoretical framework to include Biot's equations, which describe the interaction between solid deformation and pore fluid pressure. These results provide a robust foundation for future research and applications. The inclusion of Symbolic Maple routines facilitates the reproducibility of our findings,
enhancing accessibility and verification within the scientific community.

*Code availability.*    The software developed and used in this study is licensed under the MIT License. The latest version of the symbolic Maple routines is available from a permanent DOI repository (Zenodo) at: https://doi.org/10.5281/zenodo.13942953 (last accessed: 17 October 2024) (Alkhimenkov and Podladchikov, 2024). The repository contains code examples and can be readily used to reproduce the results presented in the paper. The codes are written in the Maple programming language.




## Appendix A: Explanation of the Maple Script for a single phase media

The following Maple script provides a step-by-step derivation of the entropy production for a one-dimensional system using the principles of classical non-equilibrium thermodynamics. It uses the volume-specific formulation for mass conservation and the principles of local thermodynamic equilibrium (LTE) to establish the relationship between different thermodynamic fluxes and forces. The script calculates the entropy production, $Q[s]$, and demonstrates the impact of various choices for flux definitions. Below is a detailed explanation of each step in the script.

```
1:  restart;
2:  V := 1/rho:
3:  dVdt := -diff(q[V](x), x)/rho(x): # mass balance (using volume and not density)
4:  dUdt := -diff(q[e](x), x)/rho(x): # conservation of energy
5:  dsdt := -diff(q[s](x), x)/rho(x) + Q[s]/rho(x): # balance of entropy
6:  LTE := dUdt = T(x)*dsdt + P(x)*dVdt: # local thermodynamic equilibrium
7:  Q[s] := solve(LTE, Q[s]); # solving for entropy production
8:
9:  q[e](x) := T(x)*q[s](x); # choice for energy flux
10: q[V](x) := v: # Galileo's principle for volume flux
11: q[s](x) := -lambda*diff(T(x), x): # Fourier's law for entropy flux
12: Q[s] := simplify(eval(Q[s])); # final expression for entropy production
```

**Listing 1.** Maple Script for Entropy Production

Below, we provide a detailed explanation of each line in the script.

### Initialization and Mass Conservation

```
1:  restart;
2:  V := 1/rho:
```

Here, V is defined as the specific volume, which is the inverse of density, $\rho$.

```
1:  dVdt := -diff(q[V](x), x)/rho(x):
```

This line represents the mass conservation equation using the volume-specific formulation. It calculates the time derivative of the specific volume as the negative divergence of the volume flux q[V](x) divided by the local density.

### Conservation of Energy

```
1:  dUdt := -diff(q[e](x), x)/rho(x):
```

This represents the conservation of energy, where dUdt is the time derivative of the specific internal energy, q[e](x) is the energy flux, and the equation states that the change in internal energy is equal to the negative divergence of energy flux divided by the density.

### Entropy Balance





```
1:   dsdt := -diff(q[s](x), x)/rho(x) + Q[s]/rho(x):
```

The equation represents the entropy balance. Here, `dsdt` is the time derivative of specific entropy, `q[s](x)` is the entropy flux, and `Q[s]` is the entropy production rate per unit volume. This equation states that the change in entropy is equal to the
divergence of the entropy flux plus the entropy production term.

**Local Thermodynamic Equilibrium (LTE)**

```
1:   LTE := dUdt = T(x)*dsdt + P(x)*dVdt:
```

This equation expresses the principle of local thermodynamic equilibrium (LTE). It relates the internal energy change `dUdt` to the product of temperature `T(x)` and entropy change `dsdt`, plus the product of pressure `P(x)` and the volume change `dVdt`.

**Solving for Entropy Production**

```
1:   Q[s] := solve(LTE, Q[s]);
```

The script solves the LTE equation for the entropy production term `Q[s]`.

**Choice for Energy Flux**

```
1:   q[e](x) := T(x)*q[s](x);
```

The energy flux `q[e](x)` is chosen as the product of temperature `T(x)` and the entropy flux `q[s](x)`. This is a common assumption based on the linear coupling between the energy and entropy fluxes.

**Flux Definitions**

```
1:   q[V](x) := v: # Galileo's principle for volume flux
2:   q[s](x) := -lambda*diff(T(x), x): # Fourier's law for entropy flux
```

The volume flux `q[V](x)` is represented by velocity `v` following Galileo's principle. The entropy flux `q[s](x)` is defined according to Fourier's law, where it is proportional to the temperature gradient `diff(T(x), x)` with thermal conductivity
`lambda`.

**Final Expression for Entropy Production**

```
1:   Q[s] := simplify(eval(Q[s]));
```

The final expression for entropy production `Q[s]` is simplified to:



$$Q[s] = \frac{\lambda}{T(x)} \left( \frac{dT(x)}{dx} \right)^2, \tag{A1}$$

This result shows that the entropy production is non-negative and is proportional to the square of the temperature gradient, divided by temperature, which is a classical result in non-equilibrium thermodynamics.

**Appendix B: Explanation of the Maple Script for Two-Phase Fluid-Saturated Media**

This appendix provides a detailed explanation of the Maple script used to derive the governing equations and analyze the behavior of a two-phase fluid-saturated medium. The script covers the conservation laws, flux definitions, and the derivation of entropy production for the coupled fluid and solid phases, using principles from classical non-equilibrium thermodynamics.

**General Conservation Equations**

First, we define the conservation equations for a general quantity $A(t,x)$ and mass conservation for density $\rho(t,x)$:


```
1:  restart; #some useful relations
2:  eqA := diff(rho(t, x)*A(t, x), t) + diff(rho(t, x)*A(t, x)*Vx(t, x) + qx(t, x), x) - QA;
3:  eqM := diff(rho(t, x), t) + diff(rho(t, x) * Vx(t, x), x) - Qrho;
```

- `eqA` represents the conservation of a general quantity $A(t,x)$, incorporating the advective term $\rho(t,x)A(t,x)v_x(t,x)$ and an additional flux $q_x(t,x)$. - `eqM` is the mass conservation equation for density $\rho(t,x)$ with velocity $v_x(t,x)$ and a source term $Q_\rho$. The difference between these equations is simplified to derive a general expression for the time derivative of $A(t,x)$.

```
1:  eq := simplify(eqA - eqM * A(t, x));
2:  dA_dt := solve(eq, diff(A(t, x), t));
```

The equation `eq` is derived by subtracting the mass conservation equation, multiplied by $A(t,x)$, from `eqA`. This results in an equation for the time derivative of $A(t,x)$, which is then solved to obtain `dA_dt`. Next, we calculate the total derivative of $A(t,x)$, including the convective term:

```
1:  DA_dt := collect(simplify(dA_dt + diff(A(t, x), x)*Vx(t, x)), Q);
```

The variable `DA_dt` represents the total (material) derivative of $A(t,x)$, which includes both the time derivative and the convective term $\frac{\partial A}{\partial x} \cdot v_x(t,x)$. The resulting expression is then collected and simplified with respect to the source terms $Q$:

$$\text{DA\_dt} = \frac{dA}{dt} = \frac{-A(t,x)Q_\rho - \frac{\partial q_x(t,x)}{\partial x} + Q_A}{\rho(t,x)} \tag{B1}$$

**B1 Thermodynamic Admissibility in Fluid-Saturated Porous Media**

**Simplifying Assumptions**

To simplify the model under specific assumptions, we set several parameters to zero:





```
1:   Qrhof  := 0; # No mass source or sink in the fluid phase
2:   RDarcy := 0; # Removes contribution of Qrhof from fluid momentum balance
3:   Pcor   := 0; # Allows reaction to change porosity
4:   Dc[ph] := 0; # Turns off intraphase mass diffusion
5:   eta[f] := 0; # if=0 then no full Stokes for pore scale fluid flow = only Darcy's law
6:   lam[ph]:= 0; # Turns off intraphase heat diffusion
```

**Flux Definitions and Constitutive Relations**

**Effective properties**

We define the effective properties of the solid phase using mixture rules:

1. Effective Thermal Conductivity: Starting from the total thermal conductivity:

$$\lambda_t = (1 - \phi)\lambda_s + \phi\lambda_f \tag{B2}$$

Solving for $\lambda_s$:

$$\lambda_s = \frac{\lambda_t - \lambda_f \phi}{1 - \phi} \tag{B3}$$

2. Effective Mass Diffusion Coefficient:

$$D_c^{(s)} = \frac{D_c^{(t)} - D_c^{(f)}\phi}{1 - \phi} \tag{B4}$$

3. Effective Viscosity:

$$\eta_s = \frac{\eta_t - \eta_f \phi}{1 - \phi} \tag{B5}$$

**Kinematic Relations**

```
1:   dphi_dt := diff(phi(t, x), t) + V(x) * diff(phi(t, x), x);
```

The rate of change of porosity $\phi$ is given by:

$$\frac{d\phi}{dt} = \frac{\partial\phi}{\partial t} + v\frac{\partial\phi}{\partial x} \tag{B6}$$

where $v$ is the velocity, and $\frac{d\phi}{dt}$ represents the material derivative of porosity. The fluid velocity $v^f$ relates to the solid velocity $v^s$ and the Darcy flux $q^D$:

$$v^f = v^s + \frac{q^D}{\phi}. \tag{B7}$$





**Fluxes and Source Terms**

Here we define fluxes for heat, momentum, and solute transport based on non-equilibrium thermodynamics:

```
1:  qs := -lam[ph]*phi(t, x)*diff(T(x), x) / T(x); # Fourier's law for heat flux
2:  qv := -eta[ph]*phi(t, x)*diff(V(x), x) + phi(t, x)*P(x); # Stokes' law for viscosity
3:  qc := -Dc[ph]*phi(t, x)*diff(mu(x), x); # Fick's law for diffusion
4:  qu := T(x)*qs + V(x)*qv + mu(x)*qc; # Energy flux
```

- 'qs': Heat flux defined according to Fourier's law, with thermal conductivity $\lambda[ph]$.

- 'qv': Viscous flux based on Newtonian viscosity, incorporating pressure $P(x)$.

- 'qc': Solute flux following Fick's law of diffusion, with chemical potential gradient $\mu(x)$.

- 'qu': Total energy flux, a combination of heat, mechanical, and chemical contributions.

    – Heat Flux ($q_s$). According to Fourier's law:

$$q_s = -\lambda_{\mathrm{ph}}\phi\frac{\partial T}{\partial x}\cdot\frac{1}{T}, \tag{B8}$$

    where $\lambda_{\mathrm{ph}}$ is the phase-dependent thermal conductivity.

    – Momentum Flux ($q_v$). Using Newtonian viscosity (Stokes flow approximation):

$$q_v = -\eta_{\mathrm{ph}}\phi\frac{\partial V}{\partial x} + \phi P, \tag{B9}$$

    where $\eta_{\mathrm{ph}}$ is the phase-dependent viscosity.

    – Mass Flux ($q_c$). Following Fick's law for diffusion:

$$q_c = -D_c^{(\mathrm{ph})}\phi\frac{\partial \mu}{\partial x}, \tag{B10}$$

    where $D_c^{(\mathrm{ph})}$ is the phase-dependent mass diffusion coefficient.

    – Energy Flux ($q_u$). Combining the above fluxes:

$$q_u = Tq_s + vq_v + \mu q_c \tag{B11}$$

**Balance Equations**

```
1:  qdrho_dt := (-(diff(V(x),x)*phi(t,x)+dphi_dt)*rho(t,x)       # conservation law in non-divergent form
2:                              +Qrho)/phi(t,x):
3:  dU_dt    := (-diff(qu,x) + Qu - U   *Qrho)/rho(t,x)/phi(t,x): # Energy - eq balance energy
4:  dV_dt    := (-diff(qv,x) + Qv - V(x)*Qrho)/rho(t,x)/phi(t,x): # Newton 2nd law
5:  dC_dt    := (-diff(qc,x) + Qc - C(x)*Qrho)/rho(t,x)/phi(t,x): # balance mass of solute
6:  dS_dt    := (-diff(qs,x) + Qs - S(x)*Qrho)/rho(t,x)/phi(t,x): # balance of entropy - increasing
```





– Mass Balance (Non-Divergent Form). The rate of change of density $\rho$ is:


$$\frac{d\rho}{dt} = \frac{-\left(\phi\frac{\partial v}{\partial x} + \frac{\partial \phi}{\partial t} + v\frac{\partial \phi}{\partial x}\right)\rho + Q_\rho}{\phi}, \tag{B12}$$

where $Q_\rho$ is the mass source term.

– Energy Balance

$$\frac{dU}{dt} = \frac{-\frac{\partial q_u}{\partial x} + Q_u - UQ_\rho}{\rho\phi}, \tag{B13}$$

where $Q_u$ is the energy source term.

– Momentum Balance (Newton's Second Law)

$$\frac{dv}{dt} = \frac{-\frac{\partial q_v}{\partial x} + Q_v - vQ_\rho}{\rho\phi}, \tag{B14}$$

where $Q_v$ is the momentum source term.

– Concentration Balance

$$\frac{dC}{dt} = \frac{-\frac{\partial q_c}{\partial x} + Q_c - CQ_\rho}{\rho\phi}, \tag{B15}$$

where $Q_c$ is the concentration source term.

– Entropy Balance

$$\frac{dS}{dt} = \frac{-\frac{\partial q_s}{\partial x} + Q_s - SQ_\rho}{\rho\phi}, \tag{B16}$$

where $Q_s$ is the entropy source term.

**Deriving Entropy Production**


```
1:    LET := dU_dt = T(x)*dS_dt
2:             + P(x)*drho_dt/rho(t, x)^2
3:             + V(x)*dV_dt
4:             + mu(x)*dC_dt
5:             + tau[phi]*dphie_dt/rho[ph](t,x)/(phi(t,x));
6:    TQs := simplify(T(x)*solve(LET, Qs));
```

- 'LET': The local thermodynamic equilibrium condition, which includes terms for internal energy, entropy, volume, kinetic
energy, chemical potential, and porosity change.

- 'TQs': The entropy production term, simplified from the LTE condition to ensure non-negative production.





**Local Entropy Production**

The local entropy production is derived from the energy balance and is given by:

$$\frac{dU}{dt} = T\frac{dS}{dt} + \frac{p}{\rho^2}\frac{d\rho}{dt} + v\frac{dv}{dt} + \mu\frac{dC}{dt} + \frac{\tau_\phi}{\rho\,\phi}\frac{d\phi^e}{dt}, \tag{B17}$$

where:

- $\tau_\phi$ is the thermodynamic variable (pressure) conjugated to porosity change. Note, that $\tau_\phi$ is not defined yet.
- $\dfrac{d\phi^e}{dt}$ is the reversible part of the porosity rate change.

**Physical Interpretation of Terms:**

- $T\dfrac{dS}{dt}$: Heat stored in internal energy $U$.
- $\dfrac{p}{\rho^2}\dfrac{d\rho}{dt} = -p\dfrac{d(1/\rho)}{dt}$: Work stored in elastic energy (Hooke's Law). Note that $\dfrac{dp}{K} = \dfrac{d\rho}{\rho}$, where K is the bulk modulus,
$dp = p - p_{\text{ref}}$, and $p_{\text{ref}}$ is the reference pressure.
- $v\dfrac{dv}{dt}$: Newtonian mechanics (kinetic energy, e.g., $v\frac{dv}{dt} = \frac{1}{2}\frac{dv^2}{dt}$).
- $\mu\dfrac{dC}{dt}$: Energy due to changes in composition (chemical potential), which is zero in the present derivation.
- $\dfrac{\tau_\phi}{\rho\,\phi}\dfrac{d\phi^e}{dt}$: Poroelastic effects: reversible part of the energy change due to the changes in porosity.

**Entropy Production ($TQ_s$)**

Solving the local entropy production equation for $Q_s$ and multiplying both sides by $T$, we have:

$$TQ_s = \eta\phi\left(\frac{dv}{dx}\right)^2 + \frac{\lambda\phi}{T}\left(\frac{dT}{dx}\right)^2 + pv\frac{d\phi}{dx} + \mu Q_\rho C - vQ_v - Q_\rho G_{\text{Gibbs}} + Q_u + p\frac{d\phi}{dt} - \tau_\phi\frac{d\phi^e}{dt} \tag{B18}$$

This expression represents the entropy production, which must be non-negative according to the second law of thermodynamics.

**Phase Properties and Kinematic Substitutions**

We consider both fluid and solid phases, assigning specific properties to each.

```
1:   Fluid := {ph=f,rho(t,x)=rho[f](t,x),V(x)=Vf    ,P(x)=Pf(x)       ,G(x)=Gf   ,Qv= Qvf,Qrho= Qrhof,Qc= Qcf,Qu=
         Quf,tau[phi]=0          }:
2:   Solid := {ph=s,rho(t,x)=rho[s](t,x),V(x)=Vs(x),P(x)=Pf(x)-dP(x),G(x)=Gf-dG,Qv=-Qvf,
3:   Qrho=-Qrhof,Qc=-Qcf,Qu=-Quf,phi(t,x)=1-phi(t,x)}:
4:   sbs:={diff(   phi(t,x),t) = dphife_dt+dphifvis_dt - Vs(x)*diff(   phi(t,x),x)
5:        ,diff(rho[s](t,x),t) = drhos_dt - Vs(x)*diff(rho[s](t,x),x)
6:        ,diff(rho[f](t,x),t) = drhof_dt -    Vf*diff(rho[f](t,x),x)
7:        ,diff(Vs(x),x)       = divVs};
```

We introduce substitutions for derivatives to simplify the expressions:



- Porosity Rate Change (note that porosity is divided into reversible (elastic) and irreversible (viscous) parts)

$$\frac{\partial \phi}{\partial t} = \frac{d\phi}{dt} - v^s \frac{\partial \phi}{\partial x} \tag{B19}$$

- Solid Density Rate Change

$$\frac{\partial \rho_s}{\partial t} = \frac{d\rho_s}{dt} - v^s \frac{\partial \rho_s}{\partial x} \tag{B20}$$

- Fluid Density Rate Change

$$\frac{\partial \rho_f}{\partial t} = \frac{d\rho_f}{dt} - v^f \frac{\partial \rho_f}{\partial x} \tag{B21}$$

- Solid Velocity Divergence

$$\frac{\partial v^s}{\partial x} = \operatorname{div} v^s \tag{B22}$$

**Total Entropy Production**

The entropy production for both fluid and solid phases is computed:

```
1:   TQs_total := subs(Fluid, TQs) + subs(Solid, TQs);
```

Substituting the phase properties and kinematic relations into the expression for $TQ_s$, we obtain the total entropy production:

$$TQ_{s,\text{total}} = TQ_s^{(f)} + TQ_s^{(s)} \tag{B23}$$

**Thermodynamic Variable Conjugated to Porosity Changes:**

The thermodynamic variable $\tau_\phi$ conjugated to porosity changes is now defined as:

$$\tau_\phi = \Delta p \equiv p_s - p_f. \tag{B24}$$

**Additional Relations:**

- Fluid Momentum Flux ($Q_{vf}$). Given that $Q_{\rho f} = 0$ and $R_{\text{Darcy}} = 0$:

$$Q_{vf} = \frac{\partial \phi}{\partial x} p_f - \eta_{\text{dv}} q^D \tag{B25}$$

- Porosity Rate Change in Fluid Phase ($\frac{d\phi_f}{dt}$) Since $Q_{\rho f} = 0$ and $P_{\text{cor}} = 0$:

$$\frac{d\phi_f}{dt} = \frac{d\phi^{(e)}}{dt} + k_\phi \Delta P \tag{B26}$$

- Gibbs Free Energy Change ($\Delta G$). With $P_{\text{cor}} = 0$:

$$\Delta G_{\text{Gibbs}} = \Delta G_{\text{2Gibbs}} - \frac{v^f q^D}{\phi} \tag{B27}$$

- Mass Source Term in Fluid Phase ($Q_{\rho f}$). Given by:

$$Q_{\rho f} = -k_\rho \Delta G_{\text{2Gibbs}} \tag{B28}$$

But since $Q_{\rho f} = 0$, it implies $\Delta G_{\text{2Gibbs}} = 0$ or $k_\rho = 0$.



**Total Entropy Production**

```
1:   TQs_total := collect(expand(simplify(subs(sbs, eval(TQs_total)))), {dphie_dt});
```

After simplifying and collecting terms, the total entropy production becomes:

$$TQ_{s,\text{total}} = \frac{1}{\eta_\phi}\left(\frac{p_e}{(1-\phi)}\right)^2 + \eta_t\left(\text{div}\,v^s\right)^2 + \frac{(q^D)^2\eta_{\text{dV}}}{\phi} + \frac{\lambda_t}{T}\left(\frac{\partial T}{\partial x}\right)^2. \tag{B29}$$

**As a result, entropy production is non-negative if material parameters are non-negative, which proves the thermodynamic admissibility of the two-phase system.**

**Explanation of Terms:**

- $\frac{1}{\eta_\phi}\left(\frac{p_e}{(1-\phi)}\right)^2$: Entropy production due to poroelastic deformation (poroelastic coefficient $k_\phi$ and pressure difference).
- $\eta_t\left(\text{div}\,v^s\right)^2$: Entropy production due to viscous dissipation in the solid phase.
- $\frac{(q^D)^2\eta_{\text{dV}}}{\phi}$: Entropy production due to viscous dissipation in fluid flow (Darcy flow).
- $\frac{\lambda_t}{T}\left(\frac{\partial T}{\partial x}\right)^2$: Entropy production due to heat conduction (Fourier's law).

**B2  Darcy's Law and Fluid Flow**

Darcy's law is derived for fluid flow and evaluated for the fluid phase:

```
1:   Mom_f  := 0         = subs(Fluid,  dV_dt):
2:   Mom_s  := 0         = subs(Solid,  dV_dt):
3:   qDx := simplify(solve(Mom_f, qD(x)));
```

- 'qDx': Expression for Darcy's flux, relating it to the pressure gradient. From the fluid momentum balance $\frac{dv^f}{dt} = 0$, we derive Darcy's law for the fluid flux $q^D$. Starting from the momentum balance for the fluid phase:

$$0 = \frac{-\dfrac{\partial q^v}{\partial x} + Q_{vf}}{\rho_f\phi} \tag{B30}$$

Using the expression for $q^v$ and substituting $Q_{vf}$:

$$0 = \frac{-\dfrac{\partial}{\partial x}\left(-\eta_{\text{ph}}\phi\dfrac{\partial v^f}{\partial x} + \phi p_f\right) + \left(\dfrac{\partial\phi}{\partial x}p_f - \eta_{\text{dv}}q^D\right)}{\rho_f\phi} \tag{B31}$$

Simplifying and solving for $q^D$:

$$q^D = -\frac{1}{\eta_{\text{dv}}}\frac{\partial p_f}{\partial x} \tag{B32}$$

This indicates that the fluid flux is driven by the pressure gradient and is proportional to the permeability (inverse of viscosity), which is Darcy's law.



**Solid Velocity Divergence**

Using the mass balance equations and the substitutions, we derive the divergence of the solid velocity. The mass conservation

for the solid, accounting for porosity changes, is given by:

$$\frac{\partial}{\partial t}\left(\rho_s(1-\phi)\right) + \frac{\partial}{\partial x}\left(\rho_s(1-\phi)v^s\right) = 0. \tag{B33}$$

Expanding the derivatives, we obtain:

$$(1-\phi)\frac{\partial \rho_s}{\partial t} - \rho_s\frac{\partial \phi}{\partial t} + \rho_s(1-\phi)\frac{\partial v^s}{\partial x} + v^s\frac{\partial}{\partial x}\left(\rho_s(1-\phi)\right) = 0. \tag{B34}$$

We further expand the derivative of the last term:

$$(1-\phi)\frac{\partial \rho_s}{\partial t} - \rho_s\frac{\partial \phi}{\partial t} + \rho_s(1-\phi)\frac{\partial v^s}{\partial x} + v^s(1-\phi)\frac{\partial \rho_s}{\partial x} - v^s\rho_s\frac{\partial \phi}{\partial x} = 0. \tag{B35}$$

Grouping terms and recognizing the material derivative $\frac{d^s}{dt} = \frac{\partial}{\partial t} + v^s\frac{\partial}{\partial x}$:

$$(1-\phi)\left(\frac{\partial \rho_s}{\partial t} + v^s\frac{\partial \rho_s}{\partial x}\right) - \rho_s\left(\frac{\partial \phi}{\partial t} + v^s\frac{\partial \phi}{\partial x}\right) + \rho_s(1-\phi)\frac{\partial v^s}{\partial x} = 0. \tag{B36}$$

Using the material derivatives:

$$(1-\phi)\frac{d^s\rho_s}{dt} - \rho_s\frac{d\phi}{dt} + \rho_s(1-\phi)\frac{\partial v^s}{\partial x} = 0. \tag{B37}$$

We can infer the solid velocity divergence:

$$\operatorname{div} v^s \equiv \frac{\partial v^s}{\partial x} = -\frac{1}{\rho_s}\frac{d^s\rho_s}{dt} + \frac{1}{1-\phi}\frac{d^s\phi}{dt}. \tag{B38}$$

By using equation (9) we can further simplify the expression:

$$\operatorname{div} v^s = -\frac{1}{\rho_s}\frac{d\rho_s}{dt} - \frac{1}{1-\phi}\frac{d\phi^e}{dt} - \frac{\Delta p}{\eta_\phi(1-\phi)}. \tag{B39}$$

Each term in the expression (B39) for $\operatorname{div} v^s$ has a physical interpretation:

1. Solid Density Changes:

$$-\frac{1}{\rho_s}\frac{d^s\rho_s}{dt} \tag{B40}$$

This term accounts for the volumetric changes due to variations in the solid density, such as thermal expansion or

compression under pressure.

2. Reversible Porosity Changes:

$$-\frac{1}{1-\phi}\frac{d\phi^e}{dt} \tag{B41}$$

Accounts for the reversible part of the porosity change.

3. Irreversible Porosity Changes:

$$-\frac{\Delta p}{\eta_\phi(1-\phi)} \tag{B42}$$

Incorporates the effect of pressure changes through the poroelastic coefficient $\eta_\phi$.



**Summary**

The derived expressions ensure thermodynamic admissibility by demonstrating that the total entropy production $TQ_{s,\text{total}}$ is non-negative, satisfying the second law of thermodynamics. Each term in the entropy production has a clear physical interpretation, representing the irreversible processes contributing to entropy increase in the system.

*Author contributions.* YA designed the original study, contributed to the development the symbolic routines, and wrote the manuscript. YP contributed to the early work on the derivation of Biot's and Gassmann's equations, assisted with the study design, developed the early versions of symbolic routines, helped interpret the results, edited the manuscript, and supervised the work.

*Competing interests.* The contact author has declared that none of the authors has any competing interests.

*Acknowledgements.* Yury Alkhimenkov gratefully acknowledges support from the Swiss National Science Foundation, project number
P500PN_206722.



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
