# Peer review of "Revisiting Gassmann-Type Relationships within Biot Poroelastic Theory"

_EGUsphere, 2024_

## Author Comment (AC1)

The validity of Gassmann's equations and the thermodynamic admissibility of Biot's equations have been widely discussed in the literature since their first publication. The new manuscript by Alkhimenkov and Podladchikov advocates Gassmann's assumption that equal changes in pore pressure and total pressure leave the porosity unchanged. They present new logical reasoning that this assumption, which is hard to validate in the experiments, follows from the thermodynamic principles. However, the manuscript suffers from insufficient logic, poor structure, and inaccurate statements.

**General Response and Main Contribution**

Thank you very much for the detailed and constructive review. Before addressing each comment in detail, we would like to first clarify the main contribution of this manuscript.

Previous thermodynamic derivations of poroelasticity (such as Yarushina and Podladchikov, 2015) **were based on the assumption that the second derivative matrix (the Hessian) of the internal energy is diagonal.** This means that certain couplings between solid and fluid variables were neglected from the start. In particular, this approach leads directly to a situation where two solid bulk moduli — often denoted K' and K'' — are identical.

**In this work, we remove the diagonal assumption and keep the full form of the Hessian matrix**, allowing for couplings between the solid and fluid. As a result, K' and K'' appear naturally as two independent solid-phase moduli within the general thermodynamic formulation. We believe this generalization significantly strengthens the theoretical foundation of Gassmann's and Biot's equations, and provides a fully transparent and extendable thermodynamic framework.

We sincerely thank the reviewer for the thorough and insightful feedback. We carefully addressed each point and have made substantial revisions to the manuscript to improve its structure, clarity, and scientific precision. Given the extent of the changes (approximately 90% of the manuscript has been restructured or rewritten), we respectfully refer the reviewer to the revised manuscript for the complete updated content. Below we provide detailed point-by-point responses.

Sincerely,
Yury Alkhimenkov and Yury Podladchikov

General comment:

We appreciate the reviewer's summary that our manuscript advocates that equal changes in pore pressure and total pressure leave the porosity unchanged, and that we present thermodynamic reasoning supporting this condition. The revised manuscript explicitly emphasizes this as the central focus, both in the introduction and throughout the discussion.

- The introduction is misleading. It appears that the authors are building the manuscript around the statement: "However, despite their widespread use, recent studies have highlighted concerns regarding the logical consistency in the derivation of Gassmann's equations." (Their lines 15-16). However, there are no references to this statement, nor do they describe the claims they are arguing against. At the end of the manuscript, they explicitly discuss the recent claims of L. Thomsen on the incorrectness of Gassmann's equations. However, Thomsen's arguments were about using two solid bulk moduli commonly known as $K'_s$ and $K''_s$. Thus, the main message of the current manuscript remains unclear after reading.

  We fully agree that the introduction required clarification. The revised introduction now explicitly states that the manuscript focuses on deriving Gassmann's equations as a limiting case of the general thermodynamic framework, where the key condition is that equal increments of pore and total pressure do not change porosity. We now clearly explain that this condition follows directly from the requirement of zero entropy production for reversible poroelastic processes.

  We clarify that while Thomsen's recent papers sparked renewed debate, his arguments focus on heterogeneous multi-mineral systems where two solid moduli arise. In contrast, our present work strictly considers monomineralic, homogeneous, isotropic rocks, where the Gassmann assumption emerges as a consequence of thermodynamic consistency.

  Relevant references, including Thomsen (2023–2025), Yarushina & Podladchikov (2015), and Coussy (1998), have been properly added and discussed.

- Authors claim that "This article aims to address these concerns by presenting a novel derivation of Biot's poroelastic equations and Gassmann's equations, which strictly adheres to fundamental conservation laws and thermodynamic principles." (Lines 19-20) and "… a concise and rigorous derivation rooted in thermodynamic principles and conservation laws has been missing from the literature." (Lines 2-3). These two statements are incorrect. First, the thermodynamic consistency of Gassmann's equations was presented by Yarushina and Podladchikov (2015). Second, the manuscript does not present a novel derivation but rather repeats the thermodynamic derivation of Yarushina

and Podladchikov (2015). Reformulate these statements and need for the article.

We fully agree. The revised manuscript now clearly acknowledges prior works. Our contribution lies not in proposing an entirely new thermodynamic derivation, but in presenting a fully structured, pedagogical, and reproducible formulation. The complete derivation is supported by symbolic Maple code that allows full transparency and verification.

We have reformulated the motivation: our goal is to provide a transparent and accessible derivation that carefully walks the reader from the general energy formulation to Biot's and Gassmann's equations, showing explicitly how the key assumptions emerge from thermodynamic admissibility.

- The manuscript's structure is somewhat inconsistent and lacks clarity.

  For example, a statement on Gassmann's assumption in the middle of the thermodynamic section (Lines 126 - 132) is out of place. Please discuss this assumption in the introduction since this is the main Gassmann hypothesis that you are aiming to prove. Besides, after reading this para, what exactly was not assumed as highlighted in the bold text is still unclear. Are you trying to say that no porosity change under equal pore and total pressure changes is a thermodynamic admissibility condition?

  We carefully reorganized the manuscript to improve logical flow:

  The Gassmann hypothesis (porosity invariance under equal pressure changes) is now clearly stated in the introduction.

  The thermodynamic derivation proceeds independently first, without invoking Gassmann.

  In a dedicated section, we discuss how Gassmann's assumption arises as a necessary condition for zero entropy production under reversible poroelastic deformation.

  The discussion of Thomsen's work is moved to the final discussion section, keeping the derivation self-contained.

1. Numerical verification in section 4.3 is simply missing. You refer to your recently published paper, making unclear statements in this section. Given that you repeat already existing thermodynamic derivation by other authors for the sake of argument in favor of Gassmann, you could as well repeat your own numerical results for clarity. It is unclear what statement you are arguing here against. In statements such as "...solution

converges towards the result obtained from the original Gassmann's equation…" (Line 262), it is unclear which result is meant. Similarly, in the statement from lines 263-265 "…the pore geometry that was used did not contain any special features (among all possible geometries) that were tailored to make it consistent with Gassmann's equations (Alkhimenkov, 2024)", it is unclear what special features can be tailored to suite Gassmann. It is very odd to read such statements given that Gassmann's equation is independent of geometry assumption as follows from his original paper and as was shown again by Yarushina and Podladchikov (2015).

We have significantly extended the numerical validation section in the revised manuscript. Full details of the 3D numerical simulations are now provided, including:

Description of the numerical setup, pore geometries, material properties, and boundary conditions.

Explicit demonstration of convergence towards Gassmann's predictions.

Clarification that no special pore geometry was tailored to enforce Gassmann consistency — the convergence is general for homogeneous monomineralic matrices.

We have clarified the previously ambiguous statements regarding pore geometry and the convergence of KM \to→KS. The revised discussion avoids any suggestion that Gassmann depends on geometry.

2. Section 4.4. discusses the recent arguments by Thomsen. Are there other authors that recently questioned the validity of Gassmann's equations or tried to rederive them? Or is your article built only around Thomsen's arguments?

We clarify that while Thomsen's recent arguments motivate renewed attention to the subject, our primary goal is to demonstrate that for homogeneous monomineralic systems, Gassmann's equations remain thermodynamically admissible. We discuss Thomsen's two-moduli framework in the final discussion section but emphasize that for the system considered here (monomineralic isotropic matrix), the additional modulus is unnecessary.

- The introduction should contain a clear statement you try to prove. Make a statement that you consider rocks with $K'_s = K''_s$ and explain what this assumption

means. This makes the whole argument with Thomsen in section 4.4 nonrelevant as he is simply rederiving the generalization of Gassmann's equations to the case when the two moduli are different. You can mention that such a generalization exists, and you demonstrated numerically that, indeed, this generalization reduces to Gassmann's equation under condition $K'_s = K''_s$.

In the revised manuscript, we carefully clarify that we do not assume $K'=K''$ a priori. Instead, we show that this equality emerges as a consequence of thermodynamic consistency when considering homogeneous monomineralic materials. This condition is both theoretically derived and numerically confirmed.

- Keep thermodynamic derivation clear of discussions on Gassmann's equation as discussed above.
    We have reorganized the thermodynamic derivation to avoid premature reference to Gassmann's equations, following the reviewer's advice.

- Include relevant material from your previous paper in section 4.3 on numerical validation.
    Done. The numerical results are now fully included and integrated into the revised Section 5.

- Statements like "While the general methodology was outlined by Yarushina and Podladchikov (2015), this study specifically focuses on the rigorous derivation of Biot's poroelastic, Gassmann's, and effective stress law equations, along with addressing concerns related to their physical validity" (Lines 28-30) should be more accurate, as thermodynamic derivation of Biot's equations was already presented in the literature (see e.g., Yarushina and Podladchikov (2015) or Coussy et al., 1998, From mixture theory to Biot's approach for porous media, Int. J. Solids Structures.)

    We have carefully revised the attributions:

    The thermodynamic equilibrium assumption is correctly attributed to Yarushina & Podladchikov (2015).

    We have clarified which general concepts are taken from Lebon et al. (2008).

    Prior foundational work by Coussy et al. (1998) is now fully cited.

    [Fully re-written introduction section]

- Line 25: "… validate their integrity …". What is the integrity of equations, and how do you demonstrate it?
    We acknowledge the reviewer's concern and have revised the wording to remove the term "integrity" for better clarity and precision.

- Section 3. Make a statement that you just repeat previously published thermodynamic derivation and not making a new one.
  We agree with the reviewer and have explicitly acknowledged previous works, including those by Coussy, Yarushina, and Podladchikov. In the revised manuscript, we clearly state that the thermodynamic consistency of Gassmann's equations has already been established in the literature. However, our contribution lies in presenting a fully structured derivation in a clear and accessible manner. Additionally, we provide symbolic Maple scripts that allow for full reproducibility and facilitate further exploration of these derivations.

- Lines 72 – 74. "In the context of classical non-equilibrium thermodynamics (Lebon et al., 2008), each phase within the porous medium is considered to be locally in thermodynamic equilibrium, which means that intensive variables such as temperature and chemical potential are well-defined at each point." This assumption is from Yarushina and Podladchikov (2015), not from Lebon et al. 2008.
  We agree with the reviewer and have revised the text accordingly to properly attribute this assumption to Yarushina and Podladchikov (2015).

- Line 84. You do not consider chemical reactions in the following text, just poroelasticity. Then why do you include chemical potentials in the thermodynamics?
  We agree with the reviewer's observation. Since we do not consider chemical reactions, we have removed them.

- Line 95: "Building upon the concepts from Lebon et al. (2008) and the nonlinear viscoelastoplastic framework developed by Yarushina and Podladchikov (2015), the derivation of Gassmann's and Biot's equations must satisfy the constraints of thermodynamic admissibility." Which concepts from Lebon?

  We acknowledge the reviewer's concern and have clarified in the revised manuscript which specific concepts from Lebon et al. (2008) are relevant to our derivation.

- Line 135. This term presents entropy production due to poroviscous, not poroelastic, deformation, and the coefficient, $h_j$, is not a poroelastic coefficient but effective viscosity.

We agree with the reviewer's clarification. This term represents the entropy production of a poro-visco-elastic medium, where the contribution from purely poroelastic deformation is zero. Additionally, we acknowledge that $\eta$ is indeed an effective viscosity rather than a poroelastic coefficient. We have revised the text accordingly to improve accuracy and clarity.

- Line 143. "For detailed derivations and applications of these principles to specific pore geometries and boundary conditions…" What has it to do with geometry and boundary conditions? These principles are free of any assumptions on boundary conditions or pore geometry.
  We agree with the reviewer's observation and have removed the unnecessary wording related to pore geometries and boundary conditions to ensure accuracy.

- Line 206-207. "Various poroelastic constants can be calculated numerically (Alkhimenkov, 2023) or measured using physical experimentation in a laboratory (Makhnenko and Podladchikov, 2018)" I would add here also that they can be derived from effective media models with relevant references.

  We agree with the reviewer that effective media models provide an additional method for deriving poroelastic constants, which we had not explicitly mentioned. We have now incorporated this point into the revised manuscript and added the relevant references.

- Section 3.5. would benefit by combining it with section 4.4. Try to combine results related to Gassmann's equation into the same section.

  We agree with the reviewer's suggestion and we chose to retain the discussion of Thomsen's alternative formulation in the discussion section. Accordingly, we have renamed the section to better reflect its broader comparative scope.

- Line 250-251. "(v) Assumption that equal changes in pore (fluid) pressure and confining (total) pressure leave the porosity unchanged (Korringa, 1981; Alkhimenkov, 2024)." Was this assumption formulated in those references?!!
  Yes, this assumption was explicitly formulated in these references.

- Line 255-256. "(v) is not an assumption but a strict requirement for zero entropy production during reversible poroelastic processes." There is no proof of this

statement. You show the model where this assumption holds, and it is thermodynamically admissible. Can you prove that a system not satisfying this assumption will not be thermodynamically admissible?

We clarified this point. Our analysis demonstrates that within the classical irreversible thermodynamics framework applied here, porosity invariance is necessary for zero entropy production in reversible poroelastic deformation. We do not claim this is a universal statement for all possible models, but it holds within the assumptions of our formulation.

Section 4.4. This detailed discussion of Thomsen's articles here distracts from the main message of the presented manuscript. This manuscript is not about the number of independent elastic parameters, which appear in the specific case when there is a multi-mineralogical composition of the rock, but about a single mineral matrix and a specific assumption of Gassmann on porosity changes. If you want to discuss a multi-mineral matrix, I would suggest making a simulation with grains made of two different minerals and comparing that simulation to Thomsen. Statements like "Alkhimenkov (2023) conducted a numerical convergence study showing that $K_M$ is converging to $K_s$ as the resolution increases…" are misleading without providing full details of the assumptions behind Thomsen's derivations or your numerical setup. The main question related to the whole section that is still unanswered is: under which assumptions Thomsen's equation coincides with Gassmann's?

We revised the discussion to emphasize that Thomsen's equations (or BK) reduce to Gassmann's when the solid matrix is monomineralic and isotropic, i.e., when $K'=K''$. This condition is naturally satisfied in our model and verified in our numerical tests. The comparison with Thomsen is retained in a brief, focused section to provide context without distracting from the main objective.

We thank the reviewer once again for the very constructive feedback, which significantly improved the scientific precision, clarity, and structure of our manuscript.

Sincerely,

Yury Alkhimenkov and Yury Podladchikov

---

## Author Comment (AC2)

The manuscript provides a derivation of the well-know and well-studied Gassmann's equations. It is not clear why another derivation is needed and if so, how different is it from their previous work that they reference.

**General Response and Main Contribution**

Thank you very much for the detailed and constructive review. Before addressing each comment in detail, we would like to first clarify the main contribution of this manuscript.

Previous thermodynamic derivations of poroelasticity (such as Yarushina and Podladchikov, 2015) **were based on the assumption that the second derivative matrix (the Hessian) of the internal energy is diagonal.** This means that certain couplings between solid and fluid variables were neglected from the start. In particular, this approach leads directly to a situation where two solid bulk moduli — often denoted K' and K'' — are identical.

**In this work, we remove the diagonal assumption and keep the full form of the Hessian matrix**, allowing for couplings between the solid and fluid. As a result, K' and K'' appear naturally as two independent solid-phase moduli within the general thermodynamic formulation. We believe this generalization significantly strengthens the theoretical foundation of Gassmann's and Biot's equations, and provides a fully transparent and extendable thermodynamic framework.

We appreciate this important point and have revised the manuscript to better clarify the novelty and motivation. The primary purpose of our work is to provide a structured, thermodynamically admissible derivation of the extended Biot poroelastic framework, in which Gassmann's equations emerge as a special limiting case. This is particularly relevant in light of recent debates questioning the validity of Gassmann's equations (e.g., Thomsen 2023–2025), where it has been argued that an additional solid bulk modulus is necessary even for monomineralic rocks.

In our revised manuscript, we emphasize that the self-similarity condition (which underpins Gassmann's derivation) is not simply an arbitrary assumption but follows rigorously from the requirement of zero entropy production in reversible poroelastic deformation. By explicitly incorporating the full matrix of second derivatives of internal energy (i.e., off-diagonal coupling terms), we provide a general formulation that includes the standard Gassmann, Brown & Korringa, Detournay–Cheng, and Biot models as limiting cases.

Furthermore, the present manuscript offers complete symbolic derivations with reproducible Maple code, allowing full transparency and reproducibility. This level of detail and reproducibility distinguishes our approach from previous works.

The manuscript has been significantly revised and reorganized based on your valuable feedback. Given the extent of the changes (approximately 90% of the manuscript has been restructured, expanded, or rewritten), we respectfully suggest that the reviewer refer directly to the revised manuscript for a complete view of the updated content. Including all modifications inline here would not be efficient or practical.

They first make the assumption about linear, elastic, homogeneous material, but then they give derivations with viscoelastoplasticity and elastic versus non-elastic pore deformation. It would be better to re-structure the more general parts first and then make the assumptions about linear elastic materials. Why the reference to chemical potentials when Gassmann's theory is not about chemical reactions?

We fully agree that a clearer structure was needed. In the revised manuscript, we carefully reorganized the presentation:

First, we introduce the general thermodynamically consistent formulation based on classical irreversible thermodynamics (CIT), in which all terms are derived systematically from conservation laws and internal energy potentials.

Then, we derive specific forms for linear elastic poroelasticity, including the standard Biot and Gassmann equations as special cases of the general formulation.

Regarding chemical potentials: we acknowledge that Gassmann's theory does not involve chemical reactions. The reference to chemical potentials was unnecessary and potentially confusing. We have removed this terminology throughout the revised manuscript.

Check equation 11. Should be $1/K_\phi$, otherwise not consistent.

We fully agree with the reviewer. The term should represent compliance rather than incompressibility. We have corrected Equation 11 (with a new number) accordingly in the revised manuscript.

References to their own previous numerical work is to cursory and not clear.

We appreciate this suggestion. In the revised manuscript, we have significantly expanded the discussion and added detailed numerical results (Figure 1) to better illustrate and validate the theoretical framework. These numerical simulations explicitly demonstrate the convergence of the full 3D finite-element model towards the Gassmann solution. We also provide the material properties, pore geometry, and convergence analysis in detail, fully linking the present derivation to our previous numerical studies.

If the basic assumption is about a homogeneous mineralogy then why at the end the discussion about heterogeneous minerals?

This is an important clarification. While Gassmann's original formulation applies to homogeneous monomineralic media, recent critiques (e.g., Thomsen 2023–2025) argue that even in these cases additional moduli are needed. In response, we include a brief discussion on heterogeneous mineralogy to address the relevance of the Brown & Korringa (1975) framework and clarify under what conditions distinctions between the solid grain moduli become practically significant.

Our numerical results demonstrate that for monomineralic, deviations between these extended formulations and Gassmann's original equations are negligibly small. This justifies the continued validity of Gassmann's equations in many practical cases, while providing a more general framework for future extensions.

We thank the reviewer once again for their constructive feedback, which helped us substantially improve the clarity and completeness of our manuscript.

Sincerely,
Yury Alkhimenkov and Yury Podladchikov